# Nucleic Acid Sensor-Mediated PANoptosis in Viral Infection

**DOI:** 10.3390/v16060966

**Published:** 2024-06-16

**Authors:** Lili Zhu, Zehong Qi, Huali Zhang, Nian Wang

**Affiliations:** 1Department of Pathology, Hunan Cancer Hospital, The Affiliated Cancer Hospital of Xiangya School of Medicine, Central South University, Changsha 410083, China; zhulili@hnca.org.cn; 2Department of Pathophysiology, School of Basic Medical Science, Central South University, Changsha 410083, China; 226511075@csu.edu.cn; 3Key Laboratory of Sepsis Translational Medicine of Hunan, Central South University, Changsha 410083, China; 4National Medicine Functional Experimental Teaching Center, Central South University, Changsha 410083, China

**Keywords:** viral infection, nucleic acid sensor, PANoptosis

## Abstract

Innate immunity, the first line of host defense against viral infections, recognizes viral components through different pattern-recognition receptors. Nucleic acids derived from viruses are mainly recognized by Toll-like receptors, nucleotide-binding domain leucine-rich repeat-containing receptors, absent in melanoma 2-like receptors, and cytosolic DNA sensors (e.g., Z-DNA-binding protein 1 and cyclic GMP-AMP synthase). Different types of nucleic acid sensors can recognize specific viruses due to their unique structures. PANoptosis is a unique form of inflammatory cell death pathway that is triggered by innate immune sensors and driven by caspases and receptor-interacting serine/threonine kinases through PANoptosome complexes. Nucleic acid sensors (e.g., Z-DNA-binding protein 1 and absent in melanoma 2) not only detect viruses, but also mediate PANoptosis through providing scaffold for the assembly of PANoptosomes. This review summarizes the structures of different nucleic acid sensors, discusses their roles in viral infections by driving PANoptosis, and highlights the crosstalk between different nucleic acid sensors. It also underscores the promising prospect of manipulating nucleic acid sensors as a therapeutic approach for viral infections.

## 1. Introduction

Viruses are simple microorganisms that can invade host cells and cause viral infections. In the past two decades, diverse viruses (e.g., severe acute respiratory syndrome coronavirus 1 [SARS CoV-1], SARS CoV-2 [COVID-19], Middle Eastern respiratory syndrome coronavirus [MERS-CoV], and influenza A virus [IAV]) have caused pandemics, contributing to significant global public health challenges [1]. Viral infections typically lead to a spectrum of symptoms, ranging from asymptomatic to severe diseases [2]. Although most viral infections are self-limiting, some can be life-threatening. Severe viral infections can lead to sudden death by triggering lethal cytokine storms [3,4].

The innate immune system serves as the first line of host defense against viral infections. Different viruses are initially sensed by the innate immune cells, which detect viral components through different pattern-recognition receptors (PRRs) [5,6]. To date, several families of PRRs have been identified: Toll-like receptors (TLRs), nucleotide-binding domain leucine-rich repeat-containing receptors (NLRs), absent in melanoma 2-like receptors (ALRs), retinoic acid-inducible gene I (RIG-I)-like receptors (RLRs), C-type lectin receptors (CLRs), and cytosolic DNA sensors [6,7]. Viruses of all shapes and sizes are essentially composed of a nucleic acid core, either DNA or RNA, surrounded by an outer protein coating known as capsid [8]. Nucleic acids derived from viruses are mainly detected by endosomal TLRs (e.g., TLR3, TLR7, TLR8, and TLR9), NLRs (e.g., NLR family pyrin domain containing 1 [NLRP1], and NLRP3), RLRs (e.g., RIG-I and melanoma differentiation-associated protein 5 [MDA5]), and cytosolic DNA sensors (e.g., Z-DNA-binding protein 1 [ZBP1] and cyclic GMP-AMP synthase [CGAS]). Initially, these nucleic acid sensors trigger type I interferon (IFN-I)-dependent antiviral responses and initiate diverse programmed cell death pathways in infected cells to repress virus replication and confine viral infections to neighboring cells [9,10,11]. With the persistence of infection, nucleic acid sensor-mediated cell death drives cytokine storms and increases the lethality of viral infections [12,13,14].

PANoptosis is proposed as a lytic innate immune cell death pathway that is triggered by innate immune sensors and driven by caspases and receptor-interacting serine/threonine kinases through PANoptosome complexes [15,16]. The assembly of PANoptosomes is a prominent feature of PANoptosis [17]. Classical PANoptosomes are mainly composed of three types of proteins: pathogen-associated molecular pattern (PAMP) or damage-associated molecular pattern (DAMP) sensors (e.g., ZBP1, absent in melanoma 2 [AIM2]), adapters (e.g., apoptosis-associated speck-like protein [ASC], Fas-associated via death domain [FADD]), and catalytic effectors (e.g., receptor-interacting serine/threonine kinase 1 [RIPK1], RIPK3, caspase 1 [CASP1], and CASP8) [17]. It is now established that nucleic acid sensors (e.g., ZBP1, AIM2) can recognize different viral nucleic acids through unique structural domains, triggering the assembly of PANoptosomes to engage PANoptosis [10,18,19].

This review summarizes the structures of different nucleic acid sensors, discusses their roles in viral infections by driving PANoptosis, and highlights the crosstalk between different nucleic acid sensors. It also underscores the promising prospect of manipulating nucleic acid sensors as a therapeutic approach for viral infections.

## 2. Sensing of Viral Nucleic Acids by Nucleic Acid Sensors

While mammalian cells express diverse nucleic acid sensors, they are not redundant. In the context of viral infections, viral nucleic acids are the main PAMPs that can be detected by nucleic acid sensors. Different types of nucleic acid sensors can recognize specific viruses due to their unique structures.

### 2.1. TLRs

Among the diverse array of PRRs, TLRs have garnered the most extensive research attention. Well-characterized TLRs (TLR1-9) can be categorized into two distinct groups according to their cellular localization and specific ligands [20]. The first group, exemplified by TLR1, TLR2, and TLR4, is located on the cell surface, where it plays crucial roles in recognizing PAMPs present in the cell wall components and flagellin from bacteria, yeast, and fungi [20]. In contrast, the second group of TLRs, which are located in endosomes, is specialized in detecting pathogen-derived or self-nucleic acids, with notable members including TLR3, TLR7, TLR8, and TLR9 [21]. Each TLR features a common structural framework, comprising a horseshoe-shaped extracellular domain with leucine-rich repeats (LRRs) that facilitates the recognition of ligands. Moreover, TLRs have a transmembrane domain and a cytoplasmic toll/interleukin-1 receptor (TIR) domain, which initiates downstream signaling pathways upon activation [22,23].

Endosomal TLR3, through its ectodomain, is specialized in recognizing double-stranded RNA (dsRNA), which is the replicate intermediate of all RNA viruses [24]. However, there is limited understanding of the specific structural characteristics of viral or host-derived RNA that triggers TLR3 activation. TLR7 and TLR8 share both phylogenetic and structural similarities; they are attuned to single-stranded RNA (ssRNA) from RNA viruses that enter the endosome through endocytosis [25]. More precisely, ssRNA containing poly(U) or GU-rich sequences activate TLR7 and TLR8, such as in influenza, Sendai, and Coxsackie B viruses [26,27]. Nonetheless, they activate distinct signaling pathways to promote the production of inflammatory cytokines and interferons during RNA virus infections [28,29]. In addition, species specificity has been observed in ssRNA recognition of TLR7 and TLR8. The structural differences between TLR7 and TLR8 warrant further investigation. TLR9 acts as a sensor for DNA viruses (e.g., murine cytomegalovirus [MCMV], herpes simplex virus-1 (HSV-1), HSV-2, and adenovirus) through detecting single-stranded unmethylated cytosine-phosphateguanine (CpG) motifs in their DNA [30]. Structural studies reveal that TLR9 forms a symmetric complex with CpG-DNA via its extracellular domain [31]. Therefore, further studies are required to clarify the precise details of how these endosomal TLRs distinguish their ligands.

### 2.2. NLRs

The NLR family consists of 22 human members and at least 34 mouse members [32]. They are expressed in both immune cells (e.g., lymphocytes, macrophages, and dendritic cells) and non-immune cells (e.g., epithelial cells). NLRs are categorized into several subfamilies, which are designated as NLRA, NLRB, NLRC, and NLRP [33,34]. The NLRA subfamily is represented by a single member, the MHC-II transactivator (CIITA). Similarly, the human NLRB subfamily consists of only one member, known as NLR family apoptosis inhibitory protein (NAIP). In contrast, the NLRC subfamily is more diverse, comprising six distinct members: NLRC1 (NLR family CARD domain containing 1, also known as nucleotide binding oligomerization domain containing 1 [NOD1]), NLRC2 (also known as NOD2), NLRC3, NLRC4, NLRC5, and NLR family member X1 (NLRX1) [35]. The human NLRP subfamily is the most extensive, with a total of 14 proteins ranging from NLRP1 to NLRP14. NLRs typically contain three domains: a variable N-terminal effector region that includes a caspase recruitment domain (CARD); a pyrin domain (PYD); an acidic domain; a baculoviral inhibitor of apoptosis repeats (BIRs); a centrally located nucleotide-binding domain known as NACHT, which is critical for NLR activation; and C-terminal LRRs that sense PAMPs and negatively regulate NLR activation [36,37,38]. NLR-mediated inflammasome activation is important in the host response to both PAMPs and DAMPs. Whereas C-terminal LRRs are essential for NLRs to detect viruses, NLR-dependent inflammasomes are usually activated upon viral infections by a secondary event, such as ROS generation, ion flux, or DAMP production.

NLRs capable of sensing viral nucleic acids include NLRP1, NLRP3, NLRC2, NLRC5, and NLRX1 [39]. NLRP3 recognizes a variety of RNA and DNA viruses (e.g., IAV, adenovirus, respiratory syncytial virus [RSV], human immunodeficiency virus [HIV], and vesicular stomatitis virus [VSV]) as well as DAMPs produced during viral replication, which further triggers the activation of the NLRP3 inflammasome. In fact, the NLRP3 inflammasome can be activated by a wider range of viruses than any other inflammasome identified to date, indicating its potential role as a prevalent pathway for the host to detect viral invasions. However, direct interaction with viral structures is not required for the activation of the NLRP3 inflammasome induced by viral infection, indicating that it has intricate mechanisms for recognizing viruses. Although both belong to the NLRC subfamily, NLRC2 and NLRC5 recognize different viruses. NLRC2 is primarily implicated in RNA viral infections (e.g., VSV, RSV, parainfluenza virus 3, and IAV) [8]. NLRC5 is activated in response to both RNA and DNA viruses (e.g., Sendai virus [SeV], cytomegalovirus [CMV], and VSV). However, its role may be context-dependent and species- and cell type-specific [40]. Unlike other NLR members, NLRX1 is located in the mitochondria, where it interacts with the mitochondrial antiviral signaling protein (MAVS) through its unique N-terminal X and nucleotide binding–oligomerization domains [41]. NLRX1 can bind to viral RNA by competing with dsRNA-activated protein kinase R [42].

### 2.3. RLRs

The RLR family consists of three members: RIG-I, MDA5, and laboratory of genetics and physiology 2 (LGP2) [43]. RLRs are located in the cytosol and recognize RNA virus genomes and their products. All RLRs have a central helicase domain and a carboxy-terminal domain (CTD), which work together to detect RNA [44]. RIG-I and MDA5 have two more CARDs in the amino termini. In the contexts of viral infections, RIG-I and MDA5 are activated and then undergo conformational changes, exposing and multimerizing their CARDs to allow homotypic CARD–CARD interactions with MAVS. MAVS relays the signal to TANK-binding kinase 1 (TBK1) and IκB kinase-ε (IKKε), which induce the expression of type I interferons and other genes through activating interferon regulatory factor 3 (IRF3) and IRF7 [45]. Thus, the activation of RLRs can not only result in antiviral defense but also immunopathology when the activities of RLR are uncontrolled.

### 2.4. ALRs

The ALR family comprises four members in humans (AIM2, IFN-gamma-inducible protein 16 [IFI16], pyrin and HIN domain family member 1 [PYHIN1], and myeloid cell nuclear differentiation antigen [MNDA]) and 13 members in mice [46]. They are mainly located in the cytosol and typically include a C-terminal hematopoietic, interferon-inducible, nuclear localization (HIN) domain that is responsible for recognizing pathogenic double-stranded DNA (dsDNA) and an N-terminal pyrin domain (PYD) that can interact with PYD and CARD domain containing (PYCARD). Most studies focus on AIM2, which recognizes dsDNA viruses (e.g., MCMV and vaccinia virus, but not HSV-1) [47,48]. IFI16 directly interacts with dsDNA, which is rich in stem structures from a variety of viral sources, such as MCMV, human cytomegalovirus (HCMV), and IAV. Different from AIM2, PYHIN1 and MNDA show a predominant nuclear localization due to at least one putative nuclear localization signal [49,50]. PYHIN1 can inhibit viral replication (e.g., HSV-1, human immunodeficiency virus-1 [HIV-1]) through inhibiting viral gene expression, but the underlying mechanism remains elusive [51,52].

### 2.5. Cytosolic DNA Sensors

#### 2.5.1. ZBP1

ZBP1, also known as the DNA-dependent activator of IFN regulatory factors (DAI) and DLM-1, was originally reported as an interferon-inducible tumor-associated protein. ZBP1 was identified as an innate immune activator that detects cytosolic DNAs, especially double-stranded nucleic acids that adopt Z conformation [53]. The ZBP1 protein is composed of two tandem Z-nucleic acid (Z-NA) binding domains (namely Zα1 and Zα2) at its N-terminus, two receptor-interacting protein homotypic interaction motifs, and a C-terminal signal domain [54]. The Zα domains that determine its cytosolic distribution and localization are also responsible for specific binding to viral-derived Z-NA [55]. Notably, the Zα2 domain of ZBP1 acts as a molecular switch for triggering PANoptosis upon viral infections [56]. A third DNA-binding region located next to Zα2 is responsible for binding to Z-form DNA and right-handed B-DNA [53]. All these three DNA-binding domains are critical for sensing DNA and are indispensable for the full activation of ZBP1 [57]. ZBP1 detects a variety of DNA and RNA viruses, such as IAV, HSV-1, and Zika virus (ZIKV) [58].

#### 2.5.2. CGAS

CGAS is a 60 kD DNA-sensing protein that is highly sensitive to dsDNA. It can recognize mislocated host dsDNA, such as damaged mitochondrial DNA (mtDNA) and nuclear DNA (nDNA), as well as dsDNA from pathogen sources (e.g., CMV, HSV-1, and SARS-CoV-2) [59]. The downstream signaling partner of CGAS is stimulator of interferon response CGAMP interactor 1 (STING1). The CGAS-STING1 signaling pathway is an essential part of the innate immune system. It has become a key mechanism for the induction of tandem DNA sensing and the activation of potent innate immune defense programs [60]. Structurally, the CGAS protein consists of three domains: a DNA-binding domain located at its N-terminus, a central nucleotide transferase domain, and a C-terminal domain [61]. Upon binding with dsDNA, CGAS catalyzes the synthesis of cyclic dinucleotide GMP-AMP (ultimately 2′3′-cGAMP). 2′3′-cGAMP binds and activates STING1, inducing the formation of an activated tetramer. This tetramer then translocates from the endoplasmic reticulum to the Golgi apparatus, where STING1 binds and activates TANK-binding kinase 1 (TBK1) and inhibitors of kappa B kinase (IKK), resulting in the phosphorylation of IRF3 and nuclear factor kappa B (NF-κB) [62,63].

## 3. Nucleic Acid Sensors Mediate PANoptosis in Response to Viral Infections

Upon viral infection, a spectrum of programmed cell death pathways can be activated, such as apoptosis, necroptosis, pyroptosis, and ferroptosis. Initially, these pathways serve a beneficial role by eliminating the viruses and cells infected by the viruses. However, sustained activation of these death signals can lead to the uncontrolled release of inflammatory cytokines, triggering a lethal cytokine storm. Additionally, there is crosstalk among the molecular components of different programmed cell death signaling pathways, which leads to the conceptualization of PANoptosis. PANoptosis is identified as a unique inflammatory lytic cell death pathway that is initiated by an innate immune sensor and driven by caspases and RIPKs through the assembly of PANoptosomes. Nucleic acid sensors can not only recognize nucleic acids of viral origin but also act as scaffolds to assemble PANoptosomes, thereby triggering PANoptosis.

### 3.1. ZBP1

The ZBP1 PANoptosome was the first named and studied PANoptosome [10]. Before the conceptualization of PANoptosis, it was found that ZBP1 triggers the activation of the NLRP3 inflammasome and the induction of apoptosis, necroptosis, and pyroptosis through the RIPK1–RIPK3–CASP8 axis in response to IAV infection [64]. As a nucleic acid sensor, the Zα2 and RHIM1 domain of ZBP1 can recognize and directly bind Z-RNA and/or Z-DNA derived from pathogens and host cells. Moreover, the RHIM domain mediates the recruitment of RIPK1 and RIPK3, and the Zα2 domain is indispensable for ZBP1 to regulate IAV-induced PANoptosis and NLRP3 inflammasome activation [56]. In addition to IAV, PANoptosis is induced in the macrophages infected by VSV, HSV-1, and SARS-CoV-2 [10,12,18]. Loss- and gain-of-function experiments demonstrate that these viruses promote the formation of a multiprotein complex consisting of ZBP1 (sensor); NLRP3, ASC, and CASP1 (pyroptotic proteins); FADD, CASP6, and CASP8 (apoptotic proteins); and RIPK3 (necroptotic protein), which further drives PANoptosis by activating corresponding executioners. As ZBP1 serves as a core scaffolding protein for the formation of the cell death signaling complex, this multiprotein complex is called the ZBP1 PANoptosome (Figure 1) [10].

The expression of ZBP1 is upregulated in most cases of viral infections (e.g., IAV, HIV-1, HSV-1, and ZIKA) [9,10,11]. It rapidly induces type I interferon (IFN-I) production upon sensing of the most viral genome, which provides the first line of defense against viral infections. Whereas interferon beta 1 (IFNB1)-based antiviral therapies are generally effective to treat viral infections, ZBP1-triggered inflammatory cell death limits the therapeutic efficacy of IFNB1 during coronavirus infection [12]. Thus, the species of virus and the phase of infection should be considered when evaluating the role of ZBP1 in viral infections.

### 3.2. AIM2

AIM2 is well known for sensing cytosolic dsDNA of viral, bacterial, or host damaged cellular organelles via its HIN domain [48]. During infections with HSV-1, AIM2 interacts with the MEFV innate immunity regulator (MEFV, also known as Pyrin or [TRIM20]) and ZBP1 to drive inflammatory signaling and PANoptosis [18]. Mechanistically, AIM2, MEFV, ZBP1, and ASC, along with CASP1, CASP8, FADD, RIPK1, and RIPK3, form a large multiprotein complex termed AIM2 PANoptosome (Figure 1) [18]. HSV-1 infection promotes the cooperation between MEFV and ZBP1, which further drives the assembly of AIM2 PANoptosome. AIM2 functions upstream of MEFV and ZBP1, as AIM2 deficiency completely abrogates HSV-1 infection-induced inflammatory cell death, whereas the loss of MEFV or ZBP1 only results in a partial reduction [18]. Moreover, AIM2 deficiency diminishes the protein expression of MEFV and ZBP1 in the bone marrow-derived macrophages infected by HSV-1 [18]. The underlying mechanisms by which ZBP1 and AIM2 cooperate to regulate PANoptpsis in the context of other dsDNA virus infections warrant further investigation.

## 4. Crosstalk between Different PRR-Mediated PANoptosis Mechanisms during Viral Infections

### 4.1. ZBP1 and AIM2

As previously mentioned, both ZBP1 and AIM2 can sense viral nucleic acids and serve as scaffold proteins for the assembly of PANoptosomes. In addition, ZBP1 acts as a core component of the AIM2 PANoptosome complex during HSV-1 infection, which drives PANoptosis and cytokine release. Notably, AIM2 is considered an upstream regulator of ZBP1, as AIM2 deficiency inhibits the expression of ZBP1 and abolishes inflammatory cell death [65]. These findings indicate that viruses can engage the assembly of multiple nucleic acid sensors which form distinct PANoptosomes with shared core components. Further studies are needed to clarify the underlying molecular mechanisms by which nucleic acid sensors response to different viruses.

### 4.2. ZBP1 and CGAS

While there is currently no research showing that CGAS is directly involved in PANoptosis, the CGAS-STING1 pathway is involved in ZBP1-mediated PANoptosis in the cardiomyocytes and myeloid cells treated by doxorubicin. Mechanistically, ZBP1 and CGAS form a DNA- and RHIM-dependent complex in the cytoplasm upon sensing damaged mtDNA, which promotes the assembly of ZBP1-CGAS-RIPK1-RIPK3 complexes and induces signal transducer and activator of transcription 1 (STAT1) phosphorylation and the formation of the STAT1–STAT2–interferon regulatory factor 9 (IRF9) complex (known as the ISGF3 complex) [59]. The ISGF3 complex transactivates downstream interferon-stimulated genes (ISGs) and sustains IFN-I signaling [60]. Moreover, ZBP1-activated necroptosis results in the accumulation of cytoplasmic DNA in irradiated cancer cells, which in turn autonomously activates CGAS-STING1 signaling [61]. As an ISG, the expression of ZBP1 is regulated by the CGAS-STING1-IRF3 pathway. The STING1 agonist (diABZI) triggers PANoptosis by increasing the expression of ZBP1 and promoting the assembly of the ZBP1 PANoptosome [62]. In contrast, ischemia-induced release increases the expression of ZBP1 through upregulating STING1 in cardiomyocytes, inhibiting mtDNA-induced inflammation through dampening the RIPK3-NF-κB-NLRP3 signaling pathway [63]. Therefore, ZBP1 and CGAS-STING1 can function independently; there is also a positive feedback loop between ZBP1 and CGAS-STING1 signaling (Figure 2). Future studies should address the crosstalk between ZBP1 and CGAS-STING1 in the regulation of PANoptosis during viral infections.

### 4.3. ZBP1 and NLRP3

NLRP3, a cytosolic PRR, can be activated in response to microbial infection and cytosolic danger signals [65,66,67]. Upon activation, NLRP3 assembles with the adapter protein ASC and CASP1 to form a multimeric protein called the NLRP3 inflammasome, which triggers the initiation of pyroptosis. The activation of the NLRP3 inflammasome is involved in the pathogenesis of diverse viral infections [68,69]. NLRP3, ASC, and CASP1 are also core members of different PANoptosomes, including the ZBP1, RIPK1, and NLRP12 PANoptosomes [10,15,70]. These proteins orchestrate the initiation of the activation of pyroptosis, which contributes to the activation of PANoptosis. However, the role of NLRP3 in viral infection-induced PANoptosis is enigmatic. The murine coronavirus, specifically mouse hepatitis virus (MHV), triggers the activation of the NLRP3 inflammasome and PANoptosis. While NLRP3 deficiency completely abrogates CASP1 activation and reduces virus titers following MHV infection, it results in increased apoptosis and necroptosis via CASP8 and RIPK3 [71]. In contrast, the inhibition of NLRP3 abolishes tumor necrosis factor (TNF)-induced PANoptosis of osteoblasts and promotes osteogenic differentiation [72]. These findings indicate that the role of NLRP3 in PANoptosis is context-dependent. In the context of IAV infection, ZBP1 accounts for the activation of the NLRP3 inflammasome [73]. In contrast, ZBP1 is dispensable for NLRP3 inflammasome activation in response to other RNA virus infections (e.g., VSV) [64]. Notably, ischemia induces mitochondrial damage and the release of mtDNA in cardiomyocytes, which increases the expression of ZBP1 through upregulating STING1. Thereby, mtDNA-induced inflammation is improved through suppressing the RIPK3-NF-κB-NLRP3 pathway [63]. These findings suggest that the regulation of NLRP3 by ZBP1 is context-dependent, which may depend on the type of stimuli. Further in-depth research is required to clarify the underlying mechanisms of this discrepancy.

### 4.4. NLRP12 and NLRP3

NLRP12 is also an NLR family member; it is mainly expressed in immune cells, especially neutrophils, eosinophils, and macrophages. As a cytosolic PRR, NLRP12 senses heme and PAMPs (e.g., *Yersinia pestis* and *Plasmodium chabaudi*) to drive necrotic cell death and inflammation [70,74]. Recently, NLRP12 was found to drive PANoptosis through assembling a PANoptosome complex termed the NLRP12 PANoptosome with ASC, CASP1, NLRP3, CASP8, and RIPK3 in response to heme plus PAMPs/TNF treatment [70]. Whether NLRP12-mediated PANoptosis is also involved in its antiviral effects is currently unknown. Unlike NLRP3, NLRP12 expression is decreased in human macrophages following viral infections (e.g., ZIKV, dengue virus [DENV], and Japanese encephalitis virus [JEV]); it exerts antiviral properties and interacts with NLRP3 to block the activation of the human NLRP3 inflammasome [75,76]. Considering the interaction between NLRP12 and NLRP3, as well as the role of NLRP3 in PANoptosis during viral infections, it can be predicted that PANoptosis orchestrated by NLRP12 plays a role, to varying degrees, in the context of viral infections. Nonetheless, how NLRP12 senses viral infection and interacts with NLRP3 to trigger the assembly of the NLRP12 PANoptosome needs to be clarified first.

## 5. Targeting Nucleic Acid Sensors in the Treatment of Viral Infection

Direct activation of nucleic acid sensors by synthetic nucleic acids is intractable. Specific small-molecule agonists for ZBP1 are currently unavailable [77,78]. Thus, the development of drugs that can induce Z-NA formation provides an option for targeting ZBP1. CBL0137 was recently screened out as an inducer of Z-DNA in all mammalian cell types. It potently initiates necroptosis in cancer cells via inducing Z-DNA formation and increasing the expression of ZBP1 [79,80,81]. However, the effect of CBL0137 on viral infections and its adverse reactions remain unknown. Fisetin, a kind of flavonoid mainly located in leaves and flowers, can activate ZBP1-mediated necroptosis in human ovarian cancer cells [82]. Moreover, ABT-737, a BCL2 apoptosis regulator (BCL2) family inhibitor, can activate the ZBP1-RIPK3-MLKL pathway, thus inducing apoptosis and necroptosis in bladder cancer cells [83]. Whereas these drugs are promising for activating ZBP1 in tumors, their effects on viral infections are ill defined. In addition, the side effects of these drugs should be considered due to the multi-target actions of ZBP1. Furthermore, spontaneous activation of ZBP1 can be inhibited by its interacting protein, ADAR1, suggesting that targeting these interacting proteins may provide alternative strategies to regulate ZBP1 in the treatment of viral infections [84,85]. Targeting transcription factors that control ZBP1 transcription (e.g., interferon regulating factor 1 [IRF1], IRF3, STAT1, and heat shock factor 1 [HSF1]) may be also effective for the treatment of ZBP1-associated viral infections. Nonetheless, how to precisely control the expression of ZBP1 in target cells or tissues warrants further investigation. Similarly, the effect of AIM2 antagonists or agonists on viral infections remains unknown. Despite some inhibitors of AMI2 demonstrating inhibitory effects on the activation of the AIM2 inflammasome, it remains unclear whether they also exert an inhibitory effect on AIM2-mediated PANoptosis.

## 6. Conclusions and Prospects

Nucleic acid sensors play vital roles in detecting PAMPs, particularly viral nucleic acids. These sensors trigger inflammatory and antiviral responses. Nucleic acid sensors (e.g., ZBP1 and AIM2) trigger PANoptosis through mediating the assembly of PANoptosomes during viral infections. Based on their roles in innate immunity, nucleic acid sensors have emerged as attractive therapeutic targets for the treatment of viral infections. In consideration of the immune-dependent and -independent functions of nucleic acid sensors, the development of specific agonists and antagonists for different nucleic acid sensors is of great importance. Given that direct activation of nucleic acid sensors by synthetic nucleic acids is elusive, targeted regulation of the expression of nucleic acid sensors or their interacting proteins may be a more achievable way to treat nucleic acid sensor-dependent immunopathology during viral infections.

Further studies are required to address the following questions and topics. First, how do different nucleic acid receptors recognize different types of viruses? When different nucleic acid receptors recognize the same type of virus, how do they cooperate to cope with this viral infection? Second, in addition to ZBP1 and AIM2, can other nucleic acid sensors also act as scaffold proteins for the assembly of PANoptosome during viral infections? Third, emerging evidence shows the crosstalk between different nucleic acid sensors in PANoptosis triggered by viral infections; the molecular switch by which viruses initiate the crosstalk and triggers PANoptosis needs to be elucidated. Fourth, since innate immunity is pivotal for the development of adaptive immune responses, the roles and regulatory mechanisms of nucleic acid sensors in adaptive immunity to viral infections warrant further exploration.

## Figures and Tables

**Figure 1 viruses-16-00966-f001:**
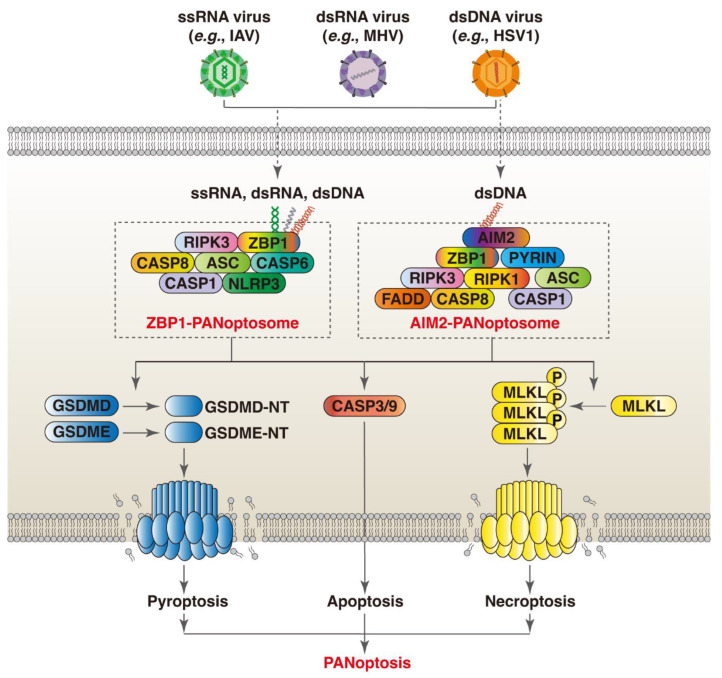
ZBP1 and AIM2 orchestrate PANoptosis in response to viral infection through assembly of PANoptosomes. (1) Nucleic acids derived from ssRNA (e.g., IAV), dsRNA (e.g., MHV), and dsDNA (e.g., HSV-1) viruses can be recognized by different nucleic acid sensors (e.g., ZBP1 and AIM2). The Zα2 domain of ZBP1 detects Z-NA of viral origin. This triggers the recruitment of RIPK1 and RIPK3 through the RHIM domain. These proteins are then linked to CASP8, and subsequently linked to NLRP3 and CASP1 via ASC to form ZBP1 PANoptosome. (2) During dsDNA virus infection (e.g., HSV-1), the HIN-200 domain of AIM2 senses virus-derived dsDNA. AIM2 dimerizes and then interact with ZBP1 via ASC to induce PANoptosis by recruiting series of death-related proteins to form AIM2 PANoptosome.

**Figure 2 viruses-16-00966-f002:**
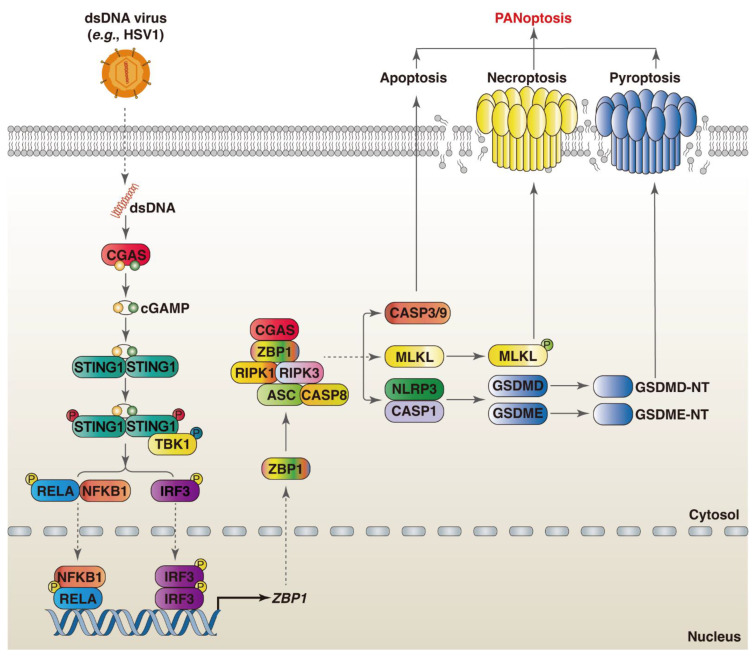
Crosstalk between ZBP1 and CGAS-mediated PANoptosis during viral infections. Upon sensing viral dsDNA, CGAS catalyzes the synthesis of 2′3′-cGAMP, which further binds and activates STING1. Thereafter, TBK1 is recruited to STING1 and phosphorylates it, resulting in the upregulation of ZBP1 expression through the activation NF-κB and IRF3. ZBP1 recruits RIPK1 and RIPK3 and induces PANoptosis through the assembly of the ZBP1 PANoptosome, activation of CASP3, and formation of MLKL oligomers and GSDMD pores.

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
