# Peer review of "Nucleic Acid Sensor-Mediated PANoptosis in Viral Infection"

_viruses, 2024, doi:10.3390/v16060966_

Round 1

Reviewer 1 Report

Comments and Suggestions for Authors

The manuscript entitled “Nucleic Acid Sensors-Mediated PANoptosis In Viral Infection” reviewed that the structures of different nucleic acid sensors, discusses their roles in viral infections by driving PANoptosis, highlights the crosstalk between different nucleic acid sensors. However, the manuscript needs major revision before it can be accepted in this journal, as explained below:

Comments 1: It is suggested that an abbreviation list can be added.

Comments 2: It is suggested tto write the full name only at the first appeared.

Comments 3: There are some detailed mistakes, so it is recommended to check the manuscript carefully.For example, the format of the fifth page should be unified.

Comments 4: It is suggested to elaborate “Crosstalk between different nucleic acid sensors mediated PANoptosis during vira infection” in depth.

Comments on the Quality of English Language

Minor editing of English language required>

Author Response

Dear reviewer:

Thanks a lot for your advice and comments concerning our research article (Manuscript Number: viruses-3012733). Those comments are all valuable and very helpful for revising and improving our manuscript, as well as the important guiding significance to our research. We have studied the comments carefully and have made corrections which we hope to meet with approval. We are pleased to respond to your comments as follows:

Comments and suggestions for authors:

The manuscript entitled “Nucleic Acid Sensors-Mediated PANoptosis In Viral Infection” reviewed that the structures of different nucleic acid sensors, discusses their roles in viral infections by driving PANoptosis, highlights the crosstalk between different nucleic acid sensors. However, the manuscript needs major revision before it can be accepted in this journal, as explained below:

--We are grateful for the referee’s comments.

Comments 1: It is suggested that an abbreviation list can be added.

--We are grateful for the referee’s comments. We have added an abbreviation list in the review.

Comments 2: It is suggested to write the full name only at the first appeared.

--We are grateful for the referee’s comments and have addressed these weaknesses accordingly.

Comments 3: There are some detailed mistakes, so it is recommended to check the manuscript carefully. For example, the format of the fifth page should be unified.

--We are grateful for the referee’s comments and have addressed these weaknesses accordingly.

Comments 4: It is suggested to elaborate “Crosstalk between different nucleic acid sensors mediated PANoptosis during viral infection” in depth.

--We are grateful for the referee’s comments and have addressed these weaknesses accordingly.

Additionally, we are glad to make a further discussion with you. We are looking forwards to your reply and your further constructive advice.

Thank you and best regards,

Sincerely yours,

Nian Wang, MD, PhD

Associate Professor,

Department of Pathophysiology,

Central South University, China

Reviewer 2 Report

Comments and Suggestions for Authors

There are excellent reviews on this topic written by the leader of this field who proposed the term of PANoptosis (e.g., JMB 434:167249, 2022). Compared to those already published, this review adds little to the literature. There are several major deficiencies that have to be rectified.

1) New pathways that have been shown to trigger PANoptosis such as RIPK1- and NLRP12-dependent pathways. These pathways and their implications in viral replication should be discussed.

2) Relevance of NLRP3 inflammasome to PANoptosis should be reviewed.

3) Controversies in the field should be highlighted and a future perspective suggesting important research directions for immediate future investigations should be provided.

Comments on the Quality of English Language

The review was generally well written.

Author Response

Dear reviewer:

Thanks a lot for your advice and comments concerning our research article (Manuscript Number: viruses-3012733). Those comments are all valuable and very helpful for revising and improving our manuscript, as well as the important guiding significance to our research. We have studied the comments carefully and have made corrections which we hope to meet with approval. We are pleased to respond to your comments as follows:

Comments and suggestions for authors:

There are excellent reviews on this topic written by the leader of this field who proposed the term of PANoptosis (e.g., JMB 434:167249, 2022). Compared to those already published, this review adds little to the literature. There are several major deficiencies that have to be rectified.

--We are grateful for the referee’s comments.

1) New pathways that have been shown to trigger PANoptosis such as RIPK1- and NLRP12-dependent pathways. These pathways and their implications in viral replication should be discussed.

--We are grateful for the referee’s comments and have addressed these weaknesses accordingly.

2) Relevance of NLRP3 inflammasome to PANoptosis should be reviewed.

--We are grateful for the referee’s comments and have addressed these weaknesses accordingly.

3) Controversies in the field should be highlighted and a future perspective suggesting important research directions for immediate future investigations should be provided.

--We are grateful for the referee’s comments and have addressed these weaknesses accordingly.

Additionally, we are glad to make a further discussion with you. We are looking forwards to your reply and your further constructive advice.

Thank you and best regards,

Sincerely yours,

Nian Wang, MD, PhD

Associate Professor,

Department of Pathophysiology,

Central South University, China

Reviewer 3 Report

Comments and Suggestions for Authors

Summary: This is a well-written review of PANoptosis.  Despite the complexity of the subject, I found the review very interesting and easy to understand. The figures and figure legends are easy to understand, and the addition of abbreviations and their definitions is appreciated. There have been several similar reviews of PANoptosis published recently (2020 to 2024). Therefore, the authors should emphasize somewhere how the information contained in this manuscript adds to the literature on this subject.

Points:

1.       The second paragraph of the Introduction is a little “choppy” and could use some transitional sentences.

2.       Line 229, typo: Probably the space between the paragraph and the next subsection head was accidentally deleted.

3.       Lines 331-332, the meaning of this sentence is unclear.  A portion may have been accidentally deleted.

4.       Line 345, should be “control” vs “controls.

Author Response

Dear reviewer:

Thanks a lot for your advice and comments concerning our research article (Manuscript Number: viruses-3012733). Those comments are all valuable and very helpful for revising and improving our manuscript, as well as the important guiding significance to our research. We have studied the comments carefully and have made corrections which we hope to meet with approval. We are pleased to respond to your comments as follows:

Comments and suggestions for authors:

This is a well-written review of PANoptosis.  Despite the complexity of the subject, I found the review very interesting and easy to understand. The figures and figure legends are easy to understand, and the addition of abbreviations and their definitions is appreciated. There have been several similar reviews of PANoptosis published recently (2020 to 2024). Therefore, the authors should emphasize somewhere how the information contained in this manuscript adds to the literature on this subject.

--We are grateful for the referee’s comments.

1. The second paragraph of the Introduction is a little “choppy” and could use some transitional sentences.

--We are grateful for the referee’s comments and have addressed these weaknesses accordingly.

2. Line 229, typo: Probably the space between the paragraph and the next subsection head was accidentally deleted.

--We are grateful for the referee’s comments and have addressed these weaknesses accordingly.

3. Lines 331-332, the meaning of this sentence is unclear. A portion may have been accidentally deleted.

--We are grateful for the referee’s comments and have addressed these weaknesses accordingly.

4. Line 345, should be “control” vs “controls.

--We are grateful for the referee’s comments and have addressed these weaknesses accordingly.

Additionally, we are glad to make a further discussion with you. We are looking forwards to your reply and your further constructive advice.

Thank you and best regards,

Sincerely yours,

Nian Wang, MD, PhD

Associate Professor,

Department of Pathophysiology,

Central South University, China

Round 2

Reviewer 2 Report

Comments and Suggestions for Authors

Some improvements have been seen in the revised version. However, it still falls short of what is required for publication in the journal. 

Comments on the Quality of English Language

Fine.

Author Response

Dear reviewer and editor:

Thanks a lot for your advice and comments concerning our research article (Manuscript number: viruses-3012733). Those comments are all valuable and very helpful for revising and improving our manuscript, as well as the important guiding significance to our research. We have studied the comments carefully and have made corrections which we hope to meet with approval. We are pleased to respond to your comments as follows:

Reviewer 2 Comments and suggestions for authors:

Comments 1: Some improvements have been seen in the revised version. However, it still falls short of what is required for publication in the journal.

--We are grateful for the reviewer’s comments. We tried our best to summarize recent progress of the roles of different nucleic acid sensor-mediated PANoptosis in viral infections and discuss their crosstalk in this review. Thus, it will enrich the understanding of the roles of nucleic acid sensors in viral infection and identify effective intervention targets.

Editor Comments and suggestions for authors:

Comments 1: The revised manuscript has now classified NLRP3 and NLRP12 as nucleic acid sensors, which is not accurate. NLRP3 is generally considered to sense changes in homeostasis that can be caused by a number of insults, and NLRP12 senses hemolytic triggers. This should be corrected.

--We are grateful for the editor’s comments. In the first round of peer review, the reviewer suggested us to discuss NLRP12-dependent PANoptosis and the relevance of NLRP3 inflammasome to PANoptosis, thus we added these two parts in 3.3 and 3.4. We have moved these two parts to 4.3, 4.4 and corrected the subheading (Crosstalk between different PRR-mediated PANoptosis during viral infections) as well as related concepts.

Comments 2: To improve accuracy and avoid confusion for readers, please ensure PANoptosis is always discussed as a distinct pathway, rather than the combined activation of apoptosis, pyroptosis, and necroptosis.

--We are grateful for the editor’s comments. We have checked the review again and ensure that PANoptosis is discussed as a distinct pathway, rather than the combined activation of apoptosis, pyroptosis, and necroptosis.

Comments 3: The review should undergo additional proofreading to correct typos and grammatical errors.

--We are grateful for the editor’s comments. We have corrected typos and grammatical errors.

Additionally, we are glad to make a further discussion with you. We are looking forwards to your reply and your further constructive advice.

Thank you and best regards,

Sincerely yours,

Nian Wang, MD, PhD

Associate Professor,

Department of Pathophysiology,

Central South University, China
